# Optimal Examination Sites for Periodontal Disease Evaluation: Applying the Item Response Theory Graded Response Model

**DOI:** 10.3390/jcm9113754

**Published:** 2020-11-21

**Authors:** Yoshiaki Nomura, Toshiya Morozumi, Mitsuo Fukuda, Nobuhiro Hanada, Erika Kakuta, Hiroaki Kobayashi, Masato Minabe, Toshiaki Nakamura, Yohei Nakayama, Fusanori Nishimura, Kazuyuki Noguchi, Yukihiro Numabe, Yorimasa Ogata, Atsushi Saito, Soh Sato, Satoshi Sekino, Naoyuki Sugano, Tsutomu Sugaya, Fumihiko Suzuki, Keiso Takahashi, Hideki Takai, Shogo Takashiba, Makoto Umeda, Hiromasa Yoshie, Atsutoshi Yoshimura, Nobuo Yoshinari, Taneaki Nakagawa

**Affiliations:** 1Department of Translational Research, Tsurumi University School of Dental Medicine, Yokohama 230-8501, Japan; nomura-y@tsurumi-u.ac.jp (Y.N.); hanada-n@tsurumi-u.ac.jp (N.H.); 2Division of Periodontology, Department of Oral Interdisciplinary Medicine, Graduate School of Dentistry, Kanagawa Dental University, Yokosuka 238-8580, Japan; minabe@kdu.ac.jp; 3Department of Periodontology, School of Dentistry, Aichi Gakuin University, Nagoya 464-8650, Japan; fukuda-m@dpc.agu.ac.jp; 4Department of Oral Microbiology, Tsurumi University School of Dental Medicine, Yokohama 230-8501, Japan; kakuta-erika@tsurumi-u.ac.jp; 5Department of Periodontology, Graduate School of Medical and Dental Sciences, Tokyo Medical and Dental University, Tokyo 113-8510, Japan; h-kobayashi.peri@tmd.ac.jp; 6Department of Periodontology, Kagoshima University Graduate School of Medical and Dental Sciences, Kagoshima 890-8544, Japan; toshi-n@dent.kagoshima-u.ac.jp (T.N.); kazuperi@dent.kagoshima-u.ac.jp (K.N.); 7Department of Periodontology, Nihon University School of Dentistry at Matsudo, Matsudo 271-8587, Japan; nakayama.youhei@nihon-u.ac.jp (Y.N.); ogata.yorimasa@nihon-u.ac.jp (Y.O.); takai.hideki@nihon-u.ac.jp (H.T.); 8Section of Periodontology, Division of Oral Rehabilitation, Faculty of Dental Science, Kyushu University, Fukuoka 819-0395, Japan; fusanori@dent.kyushu-u.ac.jp; 9Department of Periodontology, School of Life Dentistry at Tokyo, The Nippon Dental University, Tokyo 102-8159, Japan; numabe-y@tky.ndu.ac.jp (Y.N.); sekino-s@tky.ndu.ac.jp (S.S.); 10Department of Periodontology, Tokyo Dental College, Tokyo 101-0061, Japan; atsaito@tdc.ac.jp; 11Department of Periodontology, School of life Dentistry at Niigata, The Nippon Dental University, Niigata 951-8580, Japan; s-sato@ngt.ndu.ac.jp; 12Department of Periodontology, Nihon University School of Dentistry, Tokyo 101-8310, Japan; sugano.naoyuki@nihon-u.ac.jp; 13Division of Periodontology and Endodontology, Department of Oral Health Science, Hokkaido University Graduate School of Dental Medicine, Sapporo 060-8586, Japan; sugaya@den.hokudai.ac.jp; 14Division of Dental Anesthesiology, Department of Oral Surgery, Ohu University School of Dentistry, Koriyama 963-8611, Japan; f-suzuki@den.ohu-u.ac.jp; 15Division of Periodontics, Department of Conservative Dentistry, Ohu University School of Dentistry, Koriyama 963-8611, Japan; ke-takahashi@den.ohu-u.ac.jp; 16Department of Pathophysiology-Periodontal Science, Okayama University Graduate School of Medicine, Dentistry and Pharmaceutical Sciences, Okayama 700-8525, Japan; stakashi@okayama-u.ac.jp; 17Department of Periodontology, Osaka Dental University, Hirakata 573-1121, Japan; umeda-m@cc.osaka-dent.ac.jp; 18Division of Periodontology, Department of Oral Biological Science, Niigata University Graduate School of Medical and Dental Sciences, Niigata 951-8514, Japan; yoshie119@galaxy.ocn.ne.jp; 19Department of Periodontology and Endodontology, Nagasaki University Graduate School of Biomedical Sciences, Nagasaki 852-8588, Japan; ayoshi@nagasaki-u.ac.jp; 20Department of Periodontology, School of Dentistry, Matsumoto Dental University, Shiojiri 399-0781, Japan; nobuo.yoshinari@mdu.ac.jp; 21Department of Dentistry and Oral Surgery, School of Medicine, Keio University, Tokyo 60-8582, Japan; tane@z6.keio.jp

**Keywords:** periodontitis, epidemiological index, item response theory, oral examination, diagnosis, bleeding on probing

## Abstract

Periodontal examination data have a complex structure. For epidemiological studies, mass screenings, and public health use, a simple index that represents the periodontal condition is necessary. Periodontal indices for partial examination of selected teeth have been developed. However, the selected teeth vary between indices, and a justification for the selection of examination teeth has not been presented. We applied a graded response model based on the item response theory to select optimal examination teeth and sites that represent periodontal conditions. Data were obtained from 254 patients who participated in a multicenter follow-up study. Baseline data were obtained from initial follow-up. Optimal examination sites were selected using item information calculated by graded response modeling. Twelve sites—maxillary 2nd premolar (palatal-medial), 1st premolar (palatal-distal), canine (palatal-medial), lateral incisor (palatal-central), central incisor (palatal-distal) and mandibular 1st premolar (lingual, medial)—were selected. Mean values for clinical attachment level, probing pocket depth, and bleeding on probing by full mouth examinations were used for objective variables. Measuring the clinical parameters of these sites can predict the results of full mouth examination. For calculating the periodontal index by partial oral examination, a justification for the selection of examination sites is essential. This study presents an evidence-based partial examination methodology and its modeling.

## 1. Introduction

Periodontal examination should be carried out precisely for the evaluation of periodontal diseases, especially their clinical parameters. Improvement in or progression of periodontitis should be monitored at the site-level along with periodontal treatment [1]. One of the important characteristics of periodontal disease is the localization of infectious processes at specific sites, which eventually leads to tissue destruction [2]. Therefore, the accumulated data obtained during periodontal examination is copious and structurally complex [1]. Each of the 28 teeth has six examination sites for measuring parameters such as bleeding on probing (BOP), periodontal pocket probing depth (PD), and clinical attachment level (CAL). These measurements are evaluated several times throughout the course of periodontal disease treatment. Representative summary statistics include the mean value of CAL and PD, maximum value of CAL and PD, and percentage of sites with BOP (BOP%). Conventionally, summary statistics have been used in several clinical trials for the patient-level evaluation of clinical parameters [3]. However, aggregating summary statistics like the mean value or maximum value can lead to the loss of information [4]. Full-mouth protocols are proven to be the most effective [5]. When time and labor are limited, a simplified index may be necessary to represent the periodontal condition.

Several periodontal indices have been developed for epidemiological surveys and periodontal disease screening: Periodontal disease index (PDI) [6,7], Periodontal Index (PI) [8], Community periodontal index (CPI or CPITN) [8,9], Gingival bone count [10], PMA index [11], Gingival index [12]. They are based on a partial-examination method in which the target teeth for examination vary between the indices. A justification for the selection of these target teeth is not always clear. However, the developers of epidemiological indices may notice that there are several teeth or examination sites within the oral cavity that represent periodontal conditions at an individual level.

In many educational and psychological studies, a latent variable is often used as the outcome variable. These variables cannot be measured directly and are estimated by their response to the observational items. Item response theory (IRT) modeling is an important methodology commonly used for the development of tests to measure the ability by total score of test consisted of items with weighted scores [13]. Through the application of IRT, we can examine each item’s reliability and whether it contributes to an overall construct [14,15]. Total score correspond to the sum of the values of periodontal examinations. Items correspond to the values of periodontal examinations. Therefore, IRT is applicable for the indexes used in dental search. 

IRT can be applied to the clinical parameters of periodontal disease. The progress of periodontal disease at an individual level corresponds to ability. Susceptibility to the progress of PD, CAL, and BOP corresponds to item difficulty. Discrimination parameter corresponds to the predictability of each site to represent all examination sites or teeth. Previous studies have shown that IRT can efficiently characterize dental caries susceptibility [16,17]. IRT graded response modeling has also been applied in the evaluation of existing measures in several clinical areas, such as those related to swallowing and communication disorders [18,19]. Furthermore, by using IRT models with clinical diagnoses from electronic health records, a constellation of high-risk patients could be identified [20]. Therefore, by applying the IRT model to periodontal data, evidence-based target sites or teeth can be selected to represent and reflect the periodontal conditions of all sites or teeth in the oral cavity. 

This study aimed to identify the most reliable subset of teeth able to represent a full-mouth periodontal diagnosis.

## 2. Materials and Methods

### 2.1. Study Design

#### 2.1.1. Setting

This study was part of a clinical research project by the Japanese Society of Periodontology, in cooperation with 17 facilities (one clinic and 16 university hospitals) in Japan for the diagnosis of periodontitis [1,21,22]. Two-hundred-fifty-four patients with chronic periodontitis were chosen between February 2009 and February 2012 for this study, who had completed their active treatment regulated by the Japanese health insurance system. All 254 patients who registered the study were analyzed.

#### 2.1.2. Diagnosis

Each patient was diagnosed according to the guideline at the time (Guidelines of the American Academy of Periodontology) [23]. One examiner from each institute (T.M., M.F., H.K., M.M., T.N., Y.N., K.N., S.S., N.S., S.S., T.S., F.S., H.T., H.Y., A.Y., N.Y. and T.N.) was chosen to carry out the oral examinations. Each examiner was a periodontist licensed by the Japanese Society of Periodontology.

Intra- and inter-examiner calibration session were conducted at the beginning and middle of the study period. Diagnosis of periodontitis was based on the proposed criteria by the Center for Disease Control and Prevention (CDC) in partnership with the American Academy of Periodontology (AAP) [24].

#### 2.1.3. Patients

Each patient was ≥ 30 years of age, possessed at least 20 teeth, was systemically healthy, and had not been administered immunosuppressive or anti-inflammatory drugs or systemic antibiotics within 3 months before the initiation of the investigation. 

### 2.2. Research Data

In this study, we analyzed CAL, PD, BOP, plaque index (PlI), and tooth mobility. CAL was measured at six sites for all of the remaining teeth (mesiobuccal, buccal, distobuccal, mesiolingual, lingual, and distolingual). The data of CAL were categorized as < 4, 4–5, and > 5 mm.

### 2.3. Statistical Analysis

#### 2.3.1. IRT Modeling

Based on the IRT model for ordinal polytomous data, we applied a Graded Response Model [25,26,27,28,29]. Item difficulty, item discrimination, item information for the examined sites, and ability of the subjects were calculated [30,31,32,33,34]. The R software with the ltm package was used to perform the IRT analysis [27]. To reduce the total number of examination sites, sites with small item information were removed from the IRT model. This procedure were based on a step-by-step analysis. Using the data for CAL, a model was constructed for all examination sites (Model 1). Next, out of all 168 examination sites, 28 sites representing the highest information for each tooth (sum of left and right side) were selected. An IRT model was constructed using these 28 sites (Model 2). Out of these 28 sites, 12 sites in six teeth were selected for depicting a higher information (Model 3). Finally, the data from the right and left side were categorized as follows: at least one site with >5 mm CAL; at least one site with 4–5 mm CAL; or both sites with <4 mm CAL. Even though there may be optimal examination sites for each clinical parameter, the examination of numerous sites for each clinical parameter may be a laborious procedure for an epidemiological examiner or clinician. IRT models for BOP and PD were constructed in the same manner as for CAL.

#### 2.3.2. Model Evaluation

For the scatterplot, regression analysis was carried out. Generalized linear models were applied. For optimal link functions, models were evaluated using Akaike’s information criteria [35]. Receiver operating characteristic (ROC) curve was used to analyze sensitivities and specificities. The cutoff points were determined as the minimum difference between specificity and sensitivity [36,37]. The mean CAL of all examination sites and community periodontal index (CPI) were used as reference. Diagnostic criteria by the CDC-AAP [24] was used. Statistical Package for the Social Sciences version 24.0 (IBM, Tokyo, Japan) was used to perform the analyses.

To compare the model to other studies, Sensitivity, relative bias (Severity) relative bias (Extent) were calculated [38,39,40].

### 2.4. Ethical Approval

The study was conducted in compliance with the principles outlined in the Helsinki Declaration. Informed written consent was obtained from each subject, and the protocol was approved by the Institutional Review Board of each participating institution. The ethics committee members’ names and reference numbers are listed in Appendix B.

## 3. Results

### 3.1. Descriptive Statistics of the Subjects Participated in this Study

Descriptive statistics of periodontal clinical parameters were the 3.1 mm for mean of CAL, 2.5 mm for mean of PD, 15.0% for BOP%, and 0.3 for PlI. 

### 3.2. Optimal Site Selection by IRT Modeling

The final model for CAL (Model 4) is shown in Table 1, accompanied with models for the remaining clinical parameters. Item information and item response curves of Model 4 are shown in Appendix A. Using these steps, 168 examination sites were narrowed down to six variables located at 12 sites (same sites on the right and left side). The results of each step from Model 1 to 4 are shown in Appendix A. A quick reference for the calculation of ability by Model 4 is presented in Appendix C.

### 3.3. Model Evaluation

#### 3.3.1. Evaluation of Selected Sites

In clinical practice, the mean values of all examination sites are often used as summary statistics. The selected 12 sites were evaluated using a scatter plot by plotting the mean values of each clinical parameter against the mean values of the selected 12 sites. The results are shown in Figure 1. For each clinical parameter, adequate co-relations were obtained.

#### 3.3.2. Model Evaluation

The models were evaluated using two methods: correlation between predictive values and observed values and ROC curve analysis. Ability calculated by IRT analysis indicates the predictive value of the sample. The scatter plot of the ability calculated using Model 4 against the mean values of CAL is illustrated in Figure 2. A scatter plot of all 168 examination sites (Model 1) is also presented as a reference. As the plot appears to be a curve, the generalized linear model was applied. The coefficient and intercept were statistically significant. The scatter plot of the result of the generalized linear model against the models for other clinical parameters is shown in Appendix A.

The data from the same location site are combined with at least one site with >6 mm, at least one site with 4–6 mm, or both sites <4 mm.

The plot appears to be a curve. Therefore, the generalized linear model is applied for the relationship.

The selected sites were maxillary 2nd premolar (palatal-distal), maxillary 1st premolar (palatal-medial), maxillary Canine (palatal-distal), maxillary lateral incisor (palatal-central), maxillary central incisor (palatal-medial), and mandibular 1st premolar (lingual-medial).

Based on the ROC analysis, sensitivity, specificity, likelihood, and area under ROC curve (AUR) are presented in Table 2. The results of the models for other clinical parameters are also shown in Table 2. For the mean value of CAL >3 mm, sensitivity and specificity were 0.832 and 0.852, respectively, and for >5 mm, they were 0.895 and 0.911, respectively. The ROC curve of each clinical parameter and various cutoff points are presented in Appendix A. 

### 3.4. Application of the CAL Model for Diagnosis of Periodontal Disease

The cutoff point, sensitivity, specificity, and AUR for the diagnosis of periodontal disease by the CDC-AAP are presented in Table 3. ROC curves are shown in Appendix A. For moderate periodontitis, the simple mean CAL of all examination sites is most useful, followed by the mean CAL of optimal examination sites. For severe periodontitis, the CPI is most useful. Simple mean CAL were not obtained similar AUR for CPI.

### 3.5. Prediction of Conventional Periodontal Indices by the CAL Model 

The model was applied to conventionally use summary statistics, i.e., mean value of CAL, PD, and BOP%. Clinically useful cutoff points were set for each index. The results are presented in Table 4. The best predictors were the mean values of the clinical parameters by the mean values of the 12 selected sites (e.g., mean CAL by the mean CAL of the 12 sites). For obtaining the mean values of PD and BOP%, we were able to obtain higher values of the AUR compared to the CPI by calculating the simple mean CAL of the 12 selected sites. Ability of CAL, weighted CAL, could obtain higher AUR. All ROC curves are shown in Appendix A. By measuring the CAL, all other clinical parameters can be predicted.

### 3.6. Model Evaluation by Prevalence, Severity, and Extent

The prevalence, relative bias for severity and extent of model 4 were 62%, −0.0035, and −0.017, respectively.

## 4. Discussion

Public health applications of periodontal examination such as epidemiological surveys, mass screenings, and community diagnosis, simplified indices that represent an indivisible disease status are indispensable. For this purpose, several indices have been developed [6,7,8,9,10,11,12]. Developed by the World Health Organization, the CPI has been utilized in epidemiological surveys not just for community diagnosis but also for the screening of periodontal disease. Originally, the index was calculated through the examination of eight teeth [41]. It has currently been revised to include the examination of all teeth [42]; however, the original method of examining just eight teeth is also still applied. According to Japan’s Survey of Dental Diseases, national oral health surveys, which are conducted every six years, still use the original CPI examination method. However, we were unable to find any justification for the tooth-selection methodology or periodontal indices used in these surveys [43]. 

Several methods that do not require oral examinations for the screening of periodontal disease have been proposed [36,44,45,46,47,48]. These include questionnaires [49,50] and biochemical analysis of the saliva [47] or gingival crevicular fluid [51]. However, the sensitivity and specificity of questionnaires used for periodontitis screening is not high enough, and using biochemical analyses requires a special measuring device. Therefore, oral examinations are still widely used for periodontal disease screenings.

In comparison to the CPI, our model was superior in predicting clinical parameters and almost equivalent in diagnosing moderate periodontitis. Further, partial examination using the CPI requires the examination of 60 sites; our model requires the examination of only 6–12 sites. Furthermore, the examination sites presented by our model are more representative of the periodontal conditions in an oral cavity. As shown in Table 4, the partial examination of these sites represents the mean value of each examined index by the mean value of all examination sites in the oral cavity. By simply measuring the CAL, all other clinical parameters can be predicted. The model presented in this study was derived using the IRT approach. The IRT model is very useful for the selection of items that have high information. However, IRT models can only process dichotomous variables or ordinal scale; they are unable to process contentious variables. At this stage, a loss of information can occur. Therefore, the simple mean of 12 selected sites is more suitable for calculating some of the predictions presented in this study.

When the 12 selected sites are compared with other partial examination protocol, sampling sites are predominantly small. Sampling sites of other protocols were 84 [38,39,52,53], 60 [54], and 56 [38,39,52]. Partial examination protocol by high number of examination site can detect small number of deep CAL or PD. The sensitivity indicate to detect subject with at least one site of CAL > 4 mm. The sensitivity by 84 examination site was 92% [52], by 56 site was 66% [52] and by 28 site was 57% [52]. The sensitivity by the 12 site in this study was 62%. In addition, relative bias of severity of the 12 sites, which estimate the difference of mean value of CAL between full mouth examination, was −0.0035. This value protocols by 84 site were 0.009 [52], −0.046 [38] and −0.01 [53]. The 12 sites based on statistical model may equal, in other partial examination protocol, more than 5 times higher numbers of examination sites. 

In this study, the teeth selected for examination included premolars and anterior teeth; the molars were excluded. The molar is a double-rooted tooth with complex anatomical root morphology, including root length, furcation area, and divergence of root and root trunk. Cervical enamel projections and enamel pearls also occur commonly in molars and are considered to be risk factors for periodontal disease; however, their occurrence varies among different individuals [55]. Additionally, molars have to withstand high occlusal forces, which can contribute to periodontal tissue destruction. Therefore, we excluded molars from representing oral examination sites in our analysis. 

The periodontal disease index is used for assessing the periodontal status in epidemiological surveys; six target teeth (maxillary right 1stmolar, maxillary left central incisor, maxillary left 1st premolar, mandibular left 1st molar, mandibular left 1st molar, and mandibular right 1st premolar) are scored for the assessment of the disease. However, a sufficient justification for the selection of these teeth has not been provided [6]. For the index, evidence is indispensable. 

In this study, optimal sites were selected based in the item information through the models presented in Appendix A, and Table 1. Twelve sites: maxillary 2nd premolar (palatal-medial), 1st premolar (palatal-distal), 3 canine (palatal-medial), lateral incisor (palatal-central), central incisor (palatal-distal) and mandibular 1st premolar (lingual-medial) selected in this study were based on statistical modeling and represented the periodontal conditions. a full mouth examination is a best method; however, as time and labor are limited, partial examination may be applicable. Partial examination of these sites may be useful tool for epidemiological studies, mass screenings, and public health use. 

There are several limitations in this study. The study population was consisted of the patients who experienced active periodontal treatment. The wider population is necessary to confirm the robustness of the model presented in this study. However, several partial-mouth assessments were not based on the statistical modeling. The strength of the model presented in this study was based on the IRT model, and weights for the site were calculated to improve the predictive values. 

## 5. Conclusions

For calculating the periodontal index by partial oral examination, a justification for the selection of examination sites is necessary. This study presents an evidence-based partial examination method and its modeling. The 12 sites presented in this study almost equal to other partial examination protocol, which have more than 5 times the number of sampling sites.

## Figures and Tables

**Figure 1 jcm-09-03754-f001:**
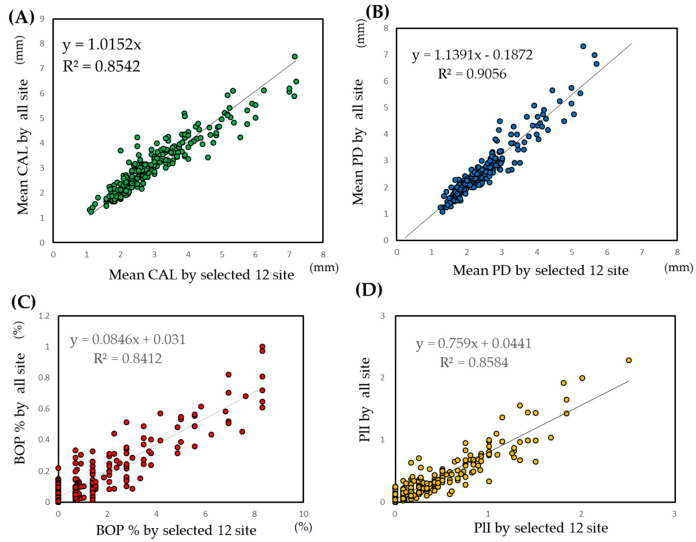
Scatter plot of the mean values of clinical parameters against the mean values of the selected 12 sites. (**A**) CAL: clinical attachment level. (**B**) PD: probing depth. (**C**) BOP: bleeding on probing. (**D**) PlI: plaque index. The selected 12 sites were the same sites that are listed in the Figure 1 legend.

**Figure 2 jcm-09-03754-f002:**
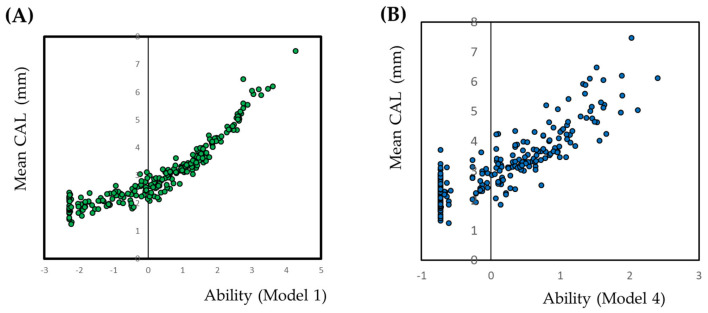
Scatter plot of the mean value of the clinical attachment level against the ability calculated by item response theory. Abilities are calculated using a graded response model under the item response theory approach. Abilities are calculated for all the 168 examined sites (**A**) and the selected six variables at the 12 sites (**B**). Ability means sum of the weighted scores of each items. In this case, ability indicate the sum of the weighted scores of positive for periodontal examination.

**Table 1 jcm-09-03754-t001:** Final model (Model 4) for the clinical attachment level.

		Maxilla					Mandibular	AIC	BIC
		2nd Premolar	1st Premolar	Canine	Lateral Incisor	Central Incisor	1st Premolar		
		Palatal	Palatal	Palatal	Palatal	Palatal	Lingual		
		Medial	Distal	Medial	Central	Distal	Medial		
CAL	Extrmt1	0.28	0.45	0.65	1.13	0.75	0.53	1796.06	1859.73
	Extrmt2	1.28	1.24	1.31	1.59	1.38	1.57		
	Discrimination	3.07	3.42	4.51	4.83	3.16	2.03		
	Item information	5.47	5.92	7.97	7.93	5.02	3.27		
PD	Extrmt1	0.72	0.88	0.96	1.35	1.01	1.03	1198.73	1262.41
	Extrmt2	1.46	1.48	1.39	1.79	1.45	1.63		
	Discrimination	3.89	3.38	4.89	4.63	3.90	3.39		
	Item information	6.82	5.37	7.87	7.41	5.94	5.40		
BOP	Extrmt1	1.30	1.22	1.56	1.45	1.63	1.95	686.60	729.04
	Discrimination	3.70	7.71	2.93	16.96	2.48	2.09		
	Item information	3.70	7.62	2.93	16.65	2.48	2.09		
PlI	Extrmt1	0.38	0.47	0.35	0.27	0.47	0.82	1965.74	2029.41
	Extrmt2	1.85	1.70	1.65	1.81	1.96	2.03		
	Discrimination	2.78	3.85	3.86	2.66	2.14	2.19		
	Item information	5.25	7.44	7.50	5.03	3.85	3.76		
Tooth Mobility	Extrmt1	0.87	0.77	1.22	0.69	0.72	1.19	1445.90	1509.57
	Extrmt2	1.90	1.85	2.21	1.79	1.99	2.56		
	Discrimination	2.41	2.53	2.98	3.81	3.23	2.07		
	Item information	4.06	4.39	5.28	7.24	6.11	3.62		

Extrmt: extremity parameters; CAL: clinical attachment level; PD: probing depth; BOP: bleeding on probing; PlI: plaque index. For CAL, it shows the cutoff to discriminate CAL < 4 mm, CAL 4−5 mm, and CAL > 5 mm. Extrmt1 discriminates CAL < 4 mm and (CAL 4−5 mm and CAL > 5 mm), and Extrmt 2 discriminates (CAL < 4 mm and CAL 4–5 mm) and CAL > 5 mm. Discrimination: This parameter shows the height of item characteristic curves. For the item response theory (IRT) analysis, CAL and PD were categorized as at least one site with >6 mm, at least one site with 4−6 mm, or both sites with <4 mm on the left or right side. IRT analysis was carried out using a graded response model. AIC: Akaike’s information criterion; BIC: Bayesian information criterion; both are fitness indices, in which small values are more suitable for a model fit.

**Table 2 jcm-09-03754-t002:** Sensitivity, specificity, and area under the receiver operating characteristic curve for the six variables.

		Cutoff Point	Sensitivity	Specificity	AUR
Mean CAL	>3 mm	−0.190	0.832	0.852	0.922
	>3.5 mm	0.193	0.865	0.878	0.939
	>4 mm	0.546	0.891	0.885	0.960
	>4.5 mm	1.033	0.929	0.929	0.978
	>5 mm	1.082	0.895	0.911	0.977
Mean PD	>3 mm	0.703	0.897	0.912	0.974
	>3.5 mm	0.891	0.926	0.930	0.983
	>4 mm	1.202	0.947	0.949	0.985
	>4.5 mm	1.391	1.000	0.971	0.989
	>5 mm	1.548	1.000	0.984	0.992
BOP	>2.5%	−0.097	0.453	0.913	0.671
	>5%	−0.097	0.388	0.939	0.698
	>10%	−0.097	0.571	0.886	0.755
	>20%	−0.097	0.687	0.834	0.804
	>25%	−0.097	0.765	0.813	0.847
PlI	>10	−0.141	0.849	0.829	0.890
	>20	0.313	0.838	0.839	0.915
	>30	0.540	0.824	0.822	0.915
	>40	0.695	0.831	0.776	0.924
	>50	0.710	0.825	0.838	0.912

CAL: clinical attachment level; PD: probing depth; BOP: bleeding on probing; PlI: plaque index; AUR: area under receiver operating characteristic curve. Cutoff points are set in abilities calculated using graded response theory. Graded response theory is one of the models of item response theory.

**Table 3 jcm-09-03754-t003:** Receiver operating characteristic analysis for the selected optimal examination sites for the diagnosis of periodontal disease.

Diagnosis	Moderate Periodontitis				Severe Periodontitis			
	Cutoff	Sensitivity	Specificity	AUR	Cutoff	Sensitivity	Specificity	AUR
CAL mean (12 sites)	2.20	0.82	0.88	0.91	2.65	0.75	0.72	0.85
CAL mean (All sites)	2.21	0.91	0.92	0.96	3.04	0.78	0.78	0.88
CAL ability (6 value)	−0.72	0.71	0.88	0.82	−0.47	0.86	0.62	0.83
CAL ability (12 sites)	−0.79	0.79	0.79	0.87	0.05	0.81	0.81	0.87
CPI	2	0.84	0.92	0.88	4	0.81	0.94	0.91

CPI: community periodontal index; CAL: clinical attachment level.

**Table 4 jcm-09-03754-t004:** Receiver operating characteristic curve analysis of the selected optimal examination sites for the mean values of pocket probing depth, clinical attachment level, and bleeding on probing.

	Cutoff	Sensitivity	Specificity	AUR	Cutoff	Sensitivity	Specificity	AUR	Cutoff	Sensitivity	Specificity	AUR
Mean CAL	Mean CAL > 3.5 mm				Mean CAL > 4 mm				Mean CAL > 4.5 mm			
CAL mean (12 sites)	3.04	0.92	0.90	0.96	3.52	0.91	0.91	0.98	3.95	0.93	0.94	0.99
CAL ability (6 values)	0.16	0.89	0.88	0.95	0.55	0.89	0.88	0.96	1.03	0.93	0.93	0.98
CAL ability (12 site)	0.35	0.89	0.90	0.96	0.69	0.93	0.93	0.97	0.90	0.93	0.93	0.99
CPI	4	0.72	0.77	0.77	2	0.96	0.36	0.81	4	0.86	0.69	0.78
Mean PD	Mean PD > 3.5 mm				Mean PD > 4 mm				Mean PD > 5 mm			
CAL mean (12 sites)	3.90	0.08	0.11	0.96	4.13	0.95	0.92	0.97	5.04	1.00	0.96	0.99
CAL ability (6 values)	4.03	0.89	0.91	0.97	0.99	0.89	0.89	0.96	1.44	1.00	0.96	0.99
CAL ability (12 sites)	0.70	0.85	0.87	0.95	0.89	0.89	0.89	0.96	1.20	1.00	0.93	0.98
CPI	4	1.00	0.71	0.85	4	1.00	0.68	0.84	4	1.00	0.65	0.82
PD mean (12 sites)	0.68	0.89	0.86	0.95	3.78	1.00	0.95	0.99	4.46	1.00	0.97	0.99
BOP	BOP > 10%				BOP > 20%				BOP > 30%			
CAL mean (12 sites)	2.65	0.64	0.65	0.71	2.96	0.76	0.75	0.83	3.26	0.82	0.80	0.89
CAL ability (6 values)	−0.26	0.65	0.62	0.69	0.12	0.74	0.77	0.80	0.46	0.77	0.82	0.86
CAL ability (12 sites)	0.03	0.63	0.66	0.67	0.31	0.74	0.75	0.80	0.60	0.75	0.85	0.85
CPI	4	0.59	0.79	0.72	4	0.75	0.76	0.80	4	0.84	0.72	0.81
BOP% mean (12 sites)	0.73	0.84	0.87	0.91	1.97	0.81	0.94	0.95	2.18	0.91	0.92	0.97

CAL: clinical attachment level; PD: probing depth; BOP: bleeding on probing; AUR: area under receiver operating characteristic curve; CPI: community periodontal index.

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
