# Peer review of "Optimal Examination Sites for Periodontal Disease Evaluation: Applying the Item Response Theory Graded Response Model"

_jcm, 2020, doi:10.3390/jcm9113754_

Round 1

Reviewer 1 Report

The manuscript from Dr. Nomura and colleagues proposed a new periodontal index by partial oral examination, which is applied a graded response model based on the item response theory. By statistical analyses using data obtained from 267 patients, the authors evidenced that the selected 12 sites could reflect the whole periodontal conditions of the oral cavity. This study is well-conducted, and the results are clear. Nevertheless, below are my concerns and recommendations for improving this manuscript.

Introduction: Although the disadvantage of conventional periodontal indices is stated adequately in the discussion part, it should be mentioned appropriately in the introduction part as well to make the authors’ study more unique.

Figures: The font size of titles and values on some graphs are too small to recognize. Please improve their visibility.

Discussion: The novelty and the future prospects in the clinical practices based on this study should be stated here. As well, it is better to mention the limitations of this study.

I hope my comments are helpful to improve the authors’ manuscript.

Author Response

Response to Reviewer 1’s Comments

Thank you for your valuable comments to improve our manuscript. We reply your comments point by point.

Introduction

Question 1): Although the disadvantage of conventional periodontal indices is stated adequately in the discussion part, it should be mentioned appropriately in the introduction part as well to make the authors’ study more unique.

Response: Thank you for your advice. One of the uniqueness of this study was selection of examination site by IRT modeling, it was included in Introduction (line 100-107).

 Figures

Question 2): The font size of titles and values on some graphs are too small to recognize. Please improve their visibility.

Response: Thank you for your indication. Figures were revised and font size was enlarged.

Discussion

Question 3): The novelty and the future prospects in the clinical practices based on this study should be stated here. As well, it is better to mention the limitations of this study.

Response: Thank you for your indication. Paragraph as strength and limitations were included in Discussion. The paragraph was highlighted in sky-blue in revised manuscript (line 345-352).

Reviewer 2 Report

This study aimed to apply a graded response model based on the item response theory to select optimal examination teeth and sites that represent periodontal conditions.

ABSTRACT

The text is confusing. Please try to rephrase it in order to create a more fluid and readable text. For example: “Mean values for clinical attachment level, probing pocket depth, bleeding on probing by examination of twelve sites” you have already mentioned the twelve sites.

INTRODUCTION

This introductory text lacks important references that demand citation. For example:

  • Owens JD, Dowsett SA, Eckert GJ, Zero DT, Kowolik MJ. Partial-mouth assessment of periodontal disease in an adult population of the United States. J Periodontol. 2003;74:1206- 13.
  • Kingman A, Susin C, Albandar JM. Effect of partial recording protocols on severity estimates of periodontal disease. J Clin Periodontol. 2008;35:659-67.
  • Kingman A, Albandar J. Methodological aspects of epidemiological studies of periodontal disease. Periodontol 2000. 2002;29:11-30.
  • Eke PI, Thornton-Evans GO, Wei L, Borgnakke WS, Dye BA. Accuracy of NHANES periodontal examination protocols. J Dent Res. 2010;89:1208-13.
  • Tran DT, Gay I, Du XL, Fu Y, Bebermeyer RD, Neumann AS, et al. Assessing periodontitis in populations: A systematic review of the validity of partial-mouth examination protocols. J Clin Periodontol. 2013;40:1064-71.
  • Tran DT, Gay I, Du XL, Fu Y, Bebermeyer RD, Neumann AS, et al. Assessment of partial-mouth periodontal examination protocols for periodontitis surveillance. J Clin Periodontol. 2014;41:846-52.
  • Kingman A, Morrison E, Löe H, Smith J. Systematic errors in estimating prevalence and severity of periodontal disease. J Periodontol. 1988;59:707-13.

What do you mean “For the evaluation of periodontal disease at an individual level, the summary statistics of these clinical parameters have been used.”?

When you state “Therefore, simplified indices are necessary to represent the periodontal condition”, I understand the latitude you want to employ to your rationale. However, to my knowledge, this is an overstatement. A simplified index may be necessary to represent the periodontal condition when time and labour are too much and we need to simplify. Otherwise, full-mouth protocols are proven to be the most effective as previously showed in Botelho et al. 2020 (doi: 10.1038/s41598-020-63700-6). Thus, I believe you should enhance this sentence.

“Several periodontal indices have been developed for epidemiological surveys and periodontal disease screening.” Which one? You failed to cite these indices. And so on in the remaining paragraph.

When you begin describing “underlying variable” I commend this transition to other fields, though you present this term for the first time and the reader may be confused. Try to explain what do you mean by “underlying variable”, put this into context.

The first paragraph of Page 2 (lines 101-121) should be reassessed. Is too confusing, and poorly objective. Try to be more concise.

“This study aimed to identify the target teeth that could represent all examination sites or teeth in the oral cavity.”, I would suggest “This study aimed to identify the most reliable subset of teeth able to represent a full-mouth periodontal diagnosis”

MATERIAL AND METHODS

The Study Design is quite unexpected. I understand that this is a multicentric study from 17 facilities. Why do you refer “In our previous reports, we had analyzed 124 subjects who successfully completed the study protocol [1, 12, 13].”? Has this subset been included in the sample?

Further, is this a prospective study? I ask this because you find it relevant to describe a set of consultations, however this study focus on diagnosis. Therefore, this information may be irrelevant.

Regarding the Diagnosis, have the examiners been calibrated? Is not enough to point the Guidelines of the American Academy of Periodontology.

Why did you decide to use the AAP 2012 case definition and not the AAP-EFP 2018 one?

In the 2.1.3 Patients section, you refer twice that patients were “systemically healthy”. Also, why did you only include patients with 30 years of age and older? Is this an inclusion criterion? If not, this should be removed and placed in the Results section.

Lines 154-155. “In this study, we analyzed CAL, PD, BOP, plaque index, and tooth mobility. Details of the data are described in our previous reports [1, 12, 13].” This is not enough nor transparent. You have to clearly describe it. I would suggest rephrasing section 2.2 section to “Periodontal measurements”.

“CAL was measured at six sites”, which sites? In order to be replicable, you have to be clear. This is one example of why you have to describe it. And the information regarding CAL should be placed in the 2.2 section.

“The R software with the ltm package was used to perform the IRT analysis.” I suggest you cite the package.

Lines 162-167, some references are lacking to support your statistical decisions.

Have you considered using a Reporting Guideline? I believe this will improve the transparency of your research.

RESULTS

Following my previous recommendation, you should start by describing your sample.

Lines 176-177. “To reduce the total number of examination sites, sites with small item information were removed from the IRT model.”. This information should be in Statistical analysis.

You refer to Models 1 to 3, though you only present Model 4 in Table 1. Are these models present in the Supplementary file? If yes, please cite them here.

Line 185. “BOP” and not “BOP%”.

Lines 185-186. “IRT models for BOP% and PD were constructed in the same manner as for CAL.” What about the Plaque Index?

Please add a better description to Figure 1 for each composite.

Line 263. I believe “AUR” should be “AUC/ROC”.

Lines 268-276. This is somehow confusing. You opted for describing CPI in here, though this information belongs to Statistics.

“The mean CAL of all examination sites and community periodontal index (CPI) were used as reference.” Why did you choose to select CPI as a reference? There is plenty of literature showing CPI as a faulty index with a high risk of misdiagnosis: 10.24873/j.rpemd.2018.11.239, 10.1038/s41598-020-63700-6, 10.1111/jcpe.12285.

Discussion

I suggest the authors rethink the Discussion section. Your first paragraph should focus on what this study provides to what is known. What adds? Why should the journal and, ultimately, the readers have to read it?

Lines 326-330. No references at all?

Lines 336-337. “Several methods that do not require oral examinations for the screening of periodontal disease have been proposed.” Do not go to common places, please cite them all here, because I do not know what methods?

What about limitations and strengths? This is mandatory.

Overall, I vividly advise a profound improvement to your Discussion.

Why haven’t you added a Conclusions section? Without it, you are not answering your research question and this is confusing.

Author Response

Response to Reviewer 2’s Comments

Thank you for your valuable comments to improve our manuscript. We reply your comments point by point. The changes according to your comments highlighted Sky-blue (Green: Reviewer 3).

Abstract

Question 1):  The text is confusing. Please try to rephrase it in order to create a more fluid and readable text. For example: “Mean values for clinical attachment level, probing pocket depth, bleeding on probing by examination of twelve sites” you have already mentioned the twelve sites.

Response: Thank you for your indication. The sentences were revised (line 66-69).

Introduction

Question 2): This introductory text lacks important references that demand citation. For example: Owens JD, Dowsett SA, Eckert GJ, Zero DT, Kowolik MJ. Partial-mouth assessment of periodontal disease in an adult population of the United States. J Periodontol. 2003;74:1206- 13.

Response: Thank you for excellent advice. By your information, we can calculate prevalence and bias. By these index we can compare other partial examination protocol. It included in Results and Discussion.

Question 3): What do you mean “For the evaluation of periodontal disease at an individual level, the summary statistics of these clinical parameters have been used.”?

 Response: Thank you for your indication. The sentence was removed in revised manuscript.

Question 4): When you state “Therefore, simplified indices are necessary to represent the periodontal condition”, I understand the latitude you want to employ to your rationale. However, to my knowledge, this is an overstatement. A simplified index may be necessary to represent the periodontal condition when time and labour are too much and we need to simplify. Otherwise, full-mouth protocols are proven to be the most effective as previously showed in Botelho et al. 2020 (doi: 10.1038/s41598-020-63700-6). Thus, I believe you should enhance this sentence.

Response: Thank you very much for your suggestion. It is precisely what we want to insisted on. Following sentence was removed and revised sentences was inserted (line 90-92).

Removed sentence: Therefore, simplified indices are necessary to represent the periodontal condition.

Inserted sentences: Full-mouth protocols are proven to be the most effective [5]. When time and labor are limited, a simplified index may be necessary to represent the periodontal condition.

New Reference: Botelho, J et al. 2020. (Ref. 5)

Question 5): “Several periodontal indices have been developed for epidemiological surveys and periodontal disease screening.” Which one? You failed to cite these indices. And so on in the remaining paragraph.

Response: Thank you for your indication. Following phrase was inserted. Periodontal disease index (PDI) [6,7], Periodontal Index (PI) [8], Community periodontal index(CPI or CPITN)[8,9], Gingival bone count[10], PMA index[11], Gingival index [12] .

Question 6): When you begin describing “underlying variable” I commend this transition to other fields, though you present this term for the first time and the reader may be confused. Try to explain what you mean by “underlying variable”, put this into context.

Response: Thank you for your indication. The paragraph was revised. Highlighted sky-blue sentences indicate the revised part (line 100-107).

 Question 7): The first paragraph of Page 2 (lines 101-121) should be reassessed. Is too confusing, and poorly objective. Try to be more concise.

Response: Thank you for your kind suggestion. The paragraph was made concise (line 108-119).

Question 8): This study aimed to identify the target teeth that could represent all examination sites or teeth in the oral cavity.”, I would suggest “This study aimed to identify the most reliable subset of teeth able to represent a full-mouth periodontal diagnosis.

Response: Thank you for presenting us of your sophisticated sentence. The sentence was replaced (line 118-119).

Materials and Methods

Question 9):  The Study Design is quite unexpected. I understand that this is a multicentric study from 17 facilities. Why do you refer “In our previous reports, we had analyzed 124 subjects who successfully completed the study protocol [1, 12, 13].”? Has this subset been included in the sample? Further, is this a prospective study? I ask this because you find it relevant to describe a set of consultations, however this study focus on diagnosis. Therefore, this information may be irrelevant.

 Response: Thank you for your indication. Following sentence was removed (line 125-133).

In our previous reports, we had analyzed 124 subjects who successfully completed the study protocol [1, 12, 13].

Regulated periodontal treatment in Japan includes: 1) periodontal examination during the first visit, 2) full-mouth supragingival scaling, 3) periodontal examination as reevaluation, 4) subgingival scaling and root planing at sites with a probing depth >4 mm or a probing depth = 4 mm with BOP, 5) periodontal examination as reevaluation and periodontal surgery if necessary, and 6) follow-up. None of the patients required periodontal surgery.

 Question 10): Regarding the Diagnosis, have the examiners been calibrated? Is not enough to point the Guidelines of the American Academy of Periodontology.

 Response: Thank you for your indication. The examiners calibrated was performed at the beginning and middle of the study period (line 139-140).

Question 11): Why did you decide to use the AAP 2012 case definition and not the AAP-EFP 2018 one?

Response: Thank you for your indication. When carried out this study, AAP-2018 was not presented. So, examiners diagnosed by AAP 2012 definitions

Question 12): In the 2.1.3 Patients section, you refer twice that patients were “systemically healthy”. Also, why did you only include patients with 30 years of age and older? Is this an inclusion criterion? If not, this should be removed and placed in the Results section.

Response: Thank you for your indication. This criteria set to exclude the juvenile periodontitis.

Question 13):  Lines 154-155. “In this study, we analyzed CAL, PD, BOP, plaque index, and tooth mobility. Details of the data are described in our previous reports [1, 12, 13]. This is not enough nor transparent. You have to clearly describe it. I would suggest rephrasing section 2.2 section to “Periodontal measurements”.

Question 14): (RESULTS) Following my previous recommendation, you should start by describing your sample.

Responses: Thank you for your indication. Following sentence was removed (line 148-149). Details of the data are described in our previous reports [1, 12, 13]. 

The data were included as descriptive statistics in Results (line 184-186).

 Question 15):  CAL was measured at six sites”, which sites? In order to be replicable, you have to be clear. This is one example of why you have to describe it. And the information regarding CAL should be placed in the 2.2 section.

 Response: Thank you for your indication. Information of CAL was moved to 2.2 section and sampling site was inserted (line 148-151).

Question 16): The R software with the ltm package was used to perform the IRT analysis.” I suggest you cite the package. Lines 162-167, some references are lacking to support your statistical decisions.

Response: Thank you for your indications. Some new references were added to the manuscript (line 152-177).

Question 17): Have you considered using a Reporting Guideline? I believe this will improve the transparency of your research.

 Response: Thank you for your indications. Structure of this manuscript was based on the STROBE guideline.

Results

Question 18): Lines 176-177. “To reduce the total number of examination sites, sites with small item information were removed from the IRT model.”. This information should be in Statistical analysis.

 Response: Thank you for your advice. Those information was moved to Statistical analysis (line 157-167).

Question 19): You refer to Models 1 to 3, though you only present Model 4 in Table 1. Are these models present in the Supplementary file? If yes, please cite them here.

Response: Thank you for your indication. It already described in previous manuscript. The results of each step from Model 1 to 4 are shown in Table S1.

Question 20): Line 185. “BOP” and not “BOP%”.

Response: Thank you for your indication. “%” was removed (line 166).

 Question 21): Lines 185-186. “IRT models for BOP% and PD were constructed in the same manner as for CAL.” What about the Plaque Index?

Response: Thank you for your indication. PlI is an index for oral hygiene. In this study, we do not construct the model for PlI.

Question 22): Please add a better description to Figure 1 for each composite.

Response: Thank you for your indication. Figures were all revised through the manuscript.

Question 23): Line 263. I believe “AUR” should be “AUC/ROC”.

 Response: Thank you for your comment. But, we think that AUR is not indicated as AUC/ROC.

Question 24): Lines 268-276. This is somehow confusing. You opted for describing CPI in here, though this information belongs to Statistics.

 Response: Thank you for your indication. The sentence moved to statistical analysis section (line 172-174).

 Question 25):  “The mean CAL of all examination sites and community periodontal index (CPI) were used as reference.” Why did you choose to select CPI as a reference? There is plenty of literature showing CPI as a faulty index with a high risk of misdiagnosis: 10.24873/j.rpemd.2018.11.239, 10.1038/s41598-020-63700-6, 10.1111/jcpe.12285.

 Response: Thank you for your indication. As you mentioned, CPI is a faulty index. However, even today, CPI are widely used in Public health. Following figure is a search results of Pubmed.

Discussion

Question 26): I suggest the authors rethink the Discussion section. Your first paragraph should focus on what this study provides to what is known. What adds? Why should the journal and, ultimately, the readers have to read it?

Response: Thank you for your indication. We removed first paragraph (line 281-288).

Question 27): Lines 326-330. No references at all?

Response: Thank you for your indication. We inserted the references (line 291).

Question 28): Lines 336-337. “Several methods that do not require oral examinations for the screening of periodontal disease have been proposed.” Do not go to common places, please cite them all here, because I do not know what methods?

Response: Thank you for your indication. We inserted the references (line 300-301).

Question 29): What about limitations and strengths? This is mandatory.

Response: Thank you for your indication. We inserted the paragraph as limitation and strength of this study (line 348-352).

Question 30): Overall, I vividly advise a profound improvement to your Discussion.

Response: Thank you for your comment. The discussion was revised by your advice including comparison other partial examination protocol and strength and limitation of this study.

Question 31): Why haven’t you added a Conclusions section? Without it, you are not answering your research question and this is confusing.

Response: Thank you for your indication. The conclusion was included (line 353-357).

Reviewer 3 Report

To authors:

This study aimed to propose a new periodontal evaluation model by applying IRT analysis and graded response model into analysis of traditional periodontal parameters. By using this new model for the prediction of some traditional periodontal clinical measurements, a faster way to diagnose patients’ periodontal condition in a quick exam of 12 sites instead of full-mouth measurements may benefit epidemiological studies, mass screenings, and public health use. The amount of clinical data from 17 dental centers is persuading. The manuscript was well written. However, there are some major revisions and clinical questions should be illustrated clearly to the readers.

  • Line 64-66:  Twelve sites: maxillary 5 (palatal, medial), 4 (palatal, distal), 3 (palatal, medial), 2 (palatal, central), 1 (palatal, distal) and mandibular 4 (lingual, medial) were selected.

Tooth numbers in this paper need to address the tooth numbering system. The same issue exists in the line 363-365.

Meanwhile, how are these 12 sites selected? Line 367-369. Any reference that supports the site selection in this study? If yes, please add reference. If no, please provide site selection reasons and evidence.

  • Line 129-139:  Please unify the descriptions of numbers in the manuscript. Example. Two-hundred-fifty-four patients vs 254 patients.
  • Line 235. Application of the graded response model with IRT was the creativity of this study. What does “ability” represent in figure 2? What does “abilities” represent and associate with clinical parameters?
  • Please unified the data analysis using the same criteria in the methodology. Example. Line 157 methodology, the groups were divided as <4 mm, <5 mm, and >5 mm. Line 195. Table. CAL <4 mm, CAL 4-6 mm, and CAL >6 mm.; Line 262. Table 2.  CAL >3mm. CAL>3.5MM, CAL>4mm, CAL>4.5mm and CAL >5mm.; Line 314. table 4. CAL>3.5mm, CAL>4mm, CAL>4.5mm
  • Line 142-144. One examiner from each institute (T.M., M.F., H.K., M.M., T.N., Y.N., K.N., S.S.,N.S., S.S., T.S., F.S., H.T., H.Y., A.Y., N.Y., and T.N.) was chosen to carry out the oral examinations. Each examiner was a periodontist licensed by the Japanese Society of Periodontology

Though each examiner is a licensed periodontist, did all examiners obtained training for this study before performance? How is the consistency of inter-rater reliability?

  • Line 314 & Line 262

How is the accuracy of the measurement of PD values at the differences of 0.5mm level?  Clinically it is very difficult to differentiate 0.5mm periodontal depth on patients.

  • Line 268. How about the result of mild periodontitis? How to accurately associate the CAL analysis result with clinical parameters such as PD, BOP and PlI? Please address the clinical application of CAL model & results with clinical indications.
  • Line 342 -346. How to apply your model by examining 6-12 sites in a clinical setting is still very confusing. Could you add an example to illustrate how to estimate or diagnose or predicate a patient's moderate periodontitis using your method?
  • Same as in line 347 “By simply measuring the CAL, all other clinical parameters can be predicted “.  Please add an example to illustrate the prediction.
  • By using your model, the most accurate application of measurement of CAL to predict all other traditional periodontal parameters is in moderate periodontitis. However, by limiting the application of the level of the periodontal condition, the clinician needs to measure at least PD and BOP to make a diagnosis of moderate periodontitis first, then apply your model. By this sequence, how could your model further improve the clinical diagnosis?
  • Line 369-370. Since this method is limited to anterior and premolars only. How could it be applied to epidemiological studies, mass screenings, and public health use where periodontal conditions of molars are common vulnerable periodontitis sites?

Author Response

Response to Reviewer 3’s Comments

Thank you for your valuable comments to improve our manuscript. We reply your comments point by point. Changes by your comments was highlighted green in revised manuscript (Sly-blue: Reviewer 2).

Question 1): Line 64-66:  Twelve sites: maxillary 5 (palatal, medial), 4 (palatal, distal), 3 (palatal, medial), 2 (palatal, central), 1 (palatal, distal) and mandibular 4 (lingual, medial) were selected. Tooth numbers in this paper need to address the tooth numbering system. The same issue exists in the line 363-365.

 Response: Thank you for your indication. Numbers of teeth were changed to tooth type through the manuscript.

Question 2): Meanwhile, how are these 12 sites selected? Line 367-369. Any reference that supports the site selection in this study? If yes, please add reference. If no, please provide site selection reasons and evidence.

 Response: Thank you for your indication. Following sentence was inserted (line 341-342).

“In this study, optimal sites were selected based in the item information through the models presented in Table S1, and Table 1.”

Question 3): Line 129-139:  Please unify the descriptions of numbers in the manuscript. Example. Two-hundred-fifty-four patients vs 254 patients.

Response: Thank you for your comment. We used properly them because Arabic number is not used when numbers come to the start of sentence.

 Question 4): Line 235. Application of the graded response model with IRT was the creativity of this study. What does “ability” represent in figure 2? What does “abilities” represent and associate with clinical parameters?

 Response: Thank you for your indication. Following sentence was inserted (line 234-235).

“Ability means sum of the weighted scores of each items. In this case, ability indicate the sum of the weighted scores of positive for periodontal examination.”

Question 5): Please unified the data analysis using the same criteria in the methodology. Example. Line 157 methodology, the groups were divided as <4 mm, <5 mm, and >5 mm. Line 195. Table. CAL <4 mm, CAL 4-6 mm, and CAL >6 mm.; Line 262. Table 2.  CAL >3mm. CAL>3.5MM, CAL>4mm, CAL>4.5mm and CAL >5mm.; Line 314. table 4. CAL>3.5mm, CAL>4mm, CAL>4.5mm

 Response: Thank you for pointing out our mistake. Groups are unified as < 4 mm, 4-5mm, and >5mm. Table 2 and 4 are diagnostic criteria, not groups.

 Question 6): Line 142-144. One examiner from each institute (T.M., M.F., H.K., M.M., T.N., Y.N., K.N., S.S.,N.S., S.S., T.S., F.S., H.T., H.Y., A.Y., N.Y., and T.N.) was chosen to carry out the oral examinations. Each examiner was a periodontist licensed by the Japanese Society of Periodontology. Though each examiner is a licensed periodontist, did all examiners obtained training for this study before performance? How is the consistency of inter-rater reliability?

 Response: The examiners calibrated was performed at the beginning and middle of the study period (line 139-140).

Question 7): Line 314 & Line 262

How is the accuracy of the measurement of PD values at the differences of 0.5mm level?  Clinically it is very difficult to differentiate 0.5mm periodontal depth on patients.

Response: Thank you for your indication. For the measurement of CAL and PD 0.5 mm levels were not set. Only the groups of <4mm, 4-5mm and 5mm<. The numbers with 0.5mm levels are mean value of PD and CAL. Mean value of CAL 4.0mm and 4.5mm may be different stage of periodontal disease.  

 Question 8): Line 268. How about the result of mild periodontitis? How to accurately associate the CAL analysis result with clinical parameters such as PD, BOP and PlI? Please address the clinical application of CAL model & results with clinical indications.

Response: Thank you for your comment. Clinical applications were inserted in discussion (line 345-347).

Question 9): Line 342 -346. How to apply your model by examining 6-12 sites in a clinical setting is still very confusing. Could you add an example to illustrate how to estimate or diagnose or predicate a patient's moderate periodontitis using your method?

Same as in line 347 “By simply measuring the CAL, all other clinical parameters can be predicted “.  Please add an example to illustrate the prediction.

Response: Thank you for your comment. Full mouth oral examination is a best method. But, performing full mouth oral examination at every time when patients visit may be almost impossible. In Japanese insurance system, patients visit dental clinics once a week or one in two weeks. Under these conditions, this model may be useful. The measurement sites were selected based on the CAL, measuring PD or BOP of the selected site can be predict the results of full mouth measurement of PD and BOP(%).

Question 10): By using your model, the most accurate application of measurement of CAL to predict all other traditional periodontal parameters is in moderate periodontitis. However, by limiting the application of the level of the periodontal condition, the clinician needs to measure at least PD and BOP to make a diagnosis of moderate periodontitis first, then apply your model. By this sequence, how could your model further improve the clinical diagnosis?

 Response: Thank you for your comment. As replied in the comments “Line 342 -346. How to apply….”, full mouth examination is a best method. For the diagnosis of periodontal disease, full mouth examination is a must. But, performing full mouth oral examination at every time when patients visit may be almost impossible.

 Question 11): Line 369-370. Since this method is limited to anterior and premolars only. How could it be applied to epidemiological studies, mass screenings, and public health use where periodontal conditions of molars are common vulnerable periodontitis sites?

 Response: Thank you for your comment. Our results seems to be inconsistent the fact that molars are vulnerable for periodontal disease. Epidemiological studies had shown that tooth loss often occurred in molar teeth. When molar teeth are lost, estimation of periodontal conditions are difficult. The teeth selected by statistical modeling were not included molar teeth. It is one of the merit of this model. The site selected were reflected the full mouth conditions. Highly affected molars may not reflect the full mouth conditions. 

Round 2

Reviewer 3 Report

The revisions improve the presentation of this manuscript to an upgraded level. The tables and figures provide sufficient information to readers. Minor english spelling check and grammar check is still needed, such as line 186. 

Author Response

Author’s response

Manuscript ID: jcm-977652

Title: Optimal Examination Sites for Periodontal Disease Evaluation: Applying the Item Response Theory Graded Response Model

Journal of Clinical Medicine

First, we would like to thank the referees and the editor for taking their valuable time to review our manuscript. The comment was helpful and constructive. Our response to the comment is listed below, and we have made changes in our manuscript accordingly.

Reviewer: 3

The revisions improve the presentation of this manuscript to an upgraded level. The tables and figures provide sufficient information to readers. Minor english spelling check and grammar check is still needed, such as line 186. 

Response: Thank you for your indication. We carefully check English spelling and grammar through the manuscript. The corrected parts were marked with highlighted yellow lines in the manuscript (line 178, 251).

We hope that the revised manuscript will now be acceptable for publication in the Journal of Clinical Medicine, and we thank you very much for your kind consideration.
